# DISSOLVING IS AMPLIFYING: TOWARDS FINE-GRAINED ANOMALY DETECTION

## ABSTRACT

In this paper, we introduce *DIA*, dissolving is amplifying. DIA is a fine-grained anomaly detection framework for medical images. We describe two novel components in the paper. First, we introduce *dissolving transformations*. Our main observation is that generative diffusion models are feature-aware and applying them to medical images in a certain manner can remove or diminish fine-grained discriminative features such as tumors or hemorrhaging. Second, we introduce an *amplifying framework* based on contrastive learning to learn a semantically meaningful representation of medical images in a self-supervised manner. The amplifying framework contrasts additional pairs of images with and without dissolving transformations applied and thereby boosts the learning of fine-grained feature representations. DIA significantly improves the medical anomaly detection performance with around 18.40% AUC boost against the baseline method and achieves an overall SOTA against other benchmark methods. Our code is available at `http://`.

## 1 INTRODUCTION

Anomaly detection aims at detecting exceptional data instances that significantly deviate from normal data. A popular application is the detection of anomalies in medical images where these anomalies often indicate a form of disease or medical problem. In the medical field, anomalous data is scarce and diverse so anomaly detection is commonly modeled as semi-supervised anomaly detection. This means that anomalous data is not available during training and the training data contains only the "normal" class.[1] Traditional anomaly detection methods include one-class methods (*e.g.* One-class SVM Chen et al. (2001)), reconstruction-based methods (*e.g.* AutoEncoders Williams et al. (2002)), and statistical models (*e.g.* HBOS Goldstein & Dengel (2012)). However, most anomaly detection methods suffer from a low recall rate meaning that many normal samples are wrongly reported as anomalies while true yet sophisticated anomalies are missed (Pang et al., 2021). Notably, due to the nature of anomalies, the collection of anomaly data can hardly cover all anomaly types even for supervised classification-based methods (Pang et al., 2019). An inherited challenge is the inconsistent behavior of anomalies, which varies without a concrete definition (Thudumu et al., 2020; Chalapathy & Chawla, 2019). Thus, identifying unseen anomalous features without requiring prior knowledge of anomalous feature patterns is crucial to anomaly detection applications.

In order to identify unseen anomalous features, many studies leveraged data augmentations (Golan & El-Yaniv, 2018; Ye et al., 2022) and adversarial features (Akcay et al., 2019b) to emphasize various feature patterns that deviate from normal data. This field attracted more attention after incorporating Generative Adversarial Networks (GANs) (Goodfellow et al., 2014), including Sabokrou et al. (2018); Ruff et al. (2018); Schlegl et al. (2017b); Akcay et al. (2019a;b); Zhao et al. (2018); Shekarizadeh et al. (2022), to enlarge the feature distances between normal and anomalous features through adversarial data generation methods. Furthermore, some studies Salem et al. (2018); Pourreza et al. (2021); Murase & Fukumizu (2022) explored the use of GANs to deconstruct images to generate out-of-distribution data for obtaining more varied anomalous features. Inspired by the recent successes of contrastive learning (Chen et al., 2020a;b; He et al., 2019; Chen et al.,

---

[1] Some early studies refer to training with only normal data as unsupervised anomaly detection. However, we follow Musa & Bouras (2021); Pang et al. (2021) and other newer methods and use the term semi-supervised.

2020c; Grill et al., 2020; Chen & He, 2021; Caron et al., 2020), contrastive-based anomaly detection methods such as Contrasting Shifted Instances (CSI) (Tack et al., 2020) and mean-shifted contrastive loss (Reiss & Hoshen, 2021) improve upon GAN-based methods by a large margin. The contrastive-based methods fit the anomaly detection context well as they are able to learn robust feature encoding without supervision. By comparing the feature differences between positive pairs (*e.g.* the same image with different views) and negative pairs (*e.g.* different images w/wo different views) without knowing the anomalous patterns, contrastive-based methods achieved outstanding performance in many general anomaly detection tasks (Tack et al., 2020; Reiss & Hoshen, 2021). However, given the low performance in experiments in Section 4, those methods are less effective for medical anomaly detection. We suspect that contrastive learning in conjunction with traditional data augmentation methods (*e.g.* crop, rotation) cannot focus on fine-grained features and only identifies coarse-grained feature differences well (*e.g.* car vs. plane). As a result, medical anomaly detection remains challenging because models struggle to recognize these fine-grained, inconspicuous yet important anomalous features that manifest differently across individual cases. These features are critical for identifying anomalies, but they can be subtle and easy to overlook. Thus, in this work, we investigate the principled question: how to emphasize the fine-grained features for fine-grained anomaly detection?

**Our method.** In this paper, we propose a new type of data augmentation that helps to learn fine-grained discriminative features. We introduce *dissolving transformations* based on pre-trained diffusion models. We observed that a clever application of a generative diffusion model can remove or suppress fine-grained discriminative features from an input image. We also introduce the framework *DIA*, dissolving is amplifying, that leverages the proposed *dissolving transformations*. DIA is a contrasting learning framework. It enhanced understanding of fine-grained discriminative features stems from a loss function that contrasts images that have been transformed with dissolving transformations to images that have not. On six medical datasets, our method obtained roughly an 18.40% AUC boost against the baseline method and achieved the overall SOTA compared to existing methods for fine-grained medical anomaly detection.

Key contributions of *DIA* include:

- **Conceptual Contribution.** We propose a novel strategy to learn fine-grained features by emphasizing the differences between images and their feature-dissolved counterparts.
- **Technical Contribution 1.** We introduce *dissolving transformations* to dissolve the fine-grained features of images. In particular, we propose *dissolving transformations* to perform semantic feature dissolving via the reverse process of generative diffusion models as described in Fig. 1.
- **Technical Contribution 2.** We propose a new *feature amplified NT-Xent loss* to learn fine-grained feature representations in a self-supervised manner. Essentially, with image pairs where only one of them has been transformed by a dissolving transformation, we compare them in a contrastive learning framework.

## 2 RELATED WORK

### 2.1 SYNTHESIS-BASED ANOMALY DETECTION

As Pang et al. (2021); Rani & E (2020) indicated, semi-supervised anomaly detection methods dominated this research field. These methods utilized only normal data whilst training. With the introduction of GANs (Goodfellow et al., 2014), many attempts have been made to bring GANs into anomaly detection. Here, we roughly categorize current methods to *positive synthesis* that increases the variation of normal data, and *negative synthesis* that generates more anomalous data.

**Positive Synthesis.** Many studies (Brock et al., 2018; Zhang et al., 2021) focused on synthesizing various in-distribution data (*i.e.* normal data) with synthetic methods. For anomaly detection tasks, earlier works such as AnoGAN (Schlegl et al., 2017a) learn normal data distributions with GANs that attempt to reconstruct the most similar images by optimizing a latent noise vector iteratively. With the success of Adversarial Auto Encoders (AAE) (Makhzani et al., 2016), some more recent studies combined AutoEncoders and GANs together to detect anomalies. GANomaly (Akcay et al., 2019a) further regularized the latent spaces between inputs and reconstructed images, then some following works improved it with more advanced generators such as UNet (Akcay et al., 2019b)

and UNet++ (Cheng et al., 2020). AnoDDPM (Wyatt et al., 2022) replaced GANs with diffusion model generators, and stated the effectiveness of noise types for medical images (i.e. Simplex noise is better than Gaussian noise). In general, most of the *positive synthesis* methods aim to improve normality feature learning despite the awareness of abnormalities, which impedes the model from understanding the anomaly feature patterns.

**Negative Synthesis.** Due to the difficulties of data acquisition and to protect patient privacy, getting high-quality, balanced datasets in medical field is difficult (Ker et al., 2017). Thus, *negative synthesis* methods are widely applied in medical image domains, such as X-ray (Salehinejad et al., 2018), lesion (Frid-Adar et al., 2018), and MRI (Han et al., 2018). Recent studies tried to integrate such negative data generation methods into anomaly detection. G2D (Pourreza et al., 2021) proposed a two-phased training to train an anomaly image generator then an anomaly detector. Similarly, AL-GAN (Murase & Fukumizu, 2022) proposed an end-to-end method that generates *pseudo-anomalies* during the training of anomaly detectors. Such GAN-based methods deconstruct images to generate *pseudo-anomalies*, resulting in unrealistic anomaly patterns, though multiple regularizers are applied to preserve image semantics. Unlike most works to synthesize novel samples from noises, we dissolve the fine-grained features on input data. Our method therefore learns the fine-grained instance feature patterns by comparing samples against their feature-dissolved counterparts. Benefiting from the step-by-step diffusing process of diffusion models, our proposed *dissolving transformations* can provide fine control over feature dissolving levels.

## 2.2 Contrastive-based Anomaly Detection

To improve anomaly detection performances, previous studies such as Dosovitskiy et al. (2014); Wen et al. (2016) explored the discriminative feature learning to reduce the needs of labelled samples for supervised anomaly detection. More recently, GeoTrans (Golan & El-Yaniv, 2018) leveraged geometric transformations to learn discriminative features, which significantly improved anomaly detection abilities. ARNet (Ye et al., 2022) attempted to use embedding-guided feature restoration to learn more semantic-preserving anomaly features. Specifically, contrastive learning methods (Chen et al., 2020a;b; He et al., 2019; Chen et al., 2020c; Grill et al., 2020; Chen & He, 2021; Caron et al., 2020) are proven to be promising in unsupervised representation learning. Inspired by the recent integration (Tack et al., 2020; Reiss & Hoshen, 2021; Cho et al., 2021) of contrastive learning and anomaly detection tasks, we propose to construct negative pairs of a given sample and its corresponding *feature-dissolved* samples in a contrastive manner to enhance the awareness of fine-grained discriminative features for medical anomaly detection.

## 3 Methodology

This section introduces DIA (Dissolving Is Amplifying), a method curated for fine-grained anomaly detection for medical imaging. DIA is a self-supervised method based on contrastive learning. DIA learns representations that can distinguish fine-grained discriminative features in medical images. First, DIA employs a dissolving strategy based on *dissolving transformations* (Section 3.1). The dissolving transformations are able to remove or deemphasize fine-grained discriminative features. Second, DIA uses the amplifying framework described in Section 3.2 to contrast images that have been transformed with and without dissolving transformations. We use the term amplifying framework as it amplifies the representation of fine-grained discriminative features.

## 3.1 Dissolving Strategy

We introduce *dissolving transformations*, a novel data augmentation strategy to create negative examples in a contrastive learning framework. The dissolving transformations can be achieved by a pre-trained diffusion model. The output image maintains a similar structure and appearance to the input image, but several fine-grained discriminative features unique to the input image are removed or attenuated. While the regular diffusion process starts with pure noise, we initialize the diffusion process with the input image without adding any noise. As depicted in Fig. 1, *dissolving transformations* gradually remove fine-grained discriminative features of various datasets (Figs. 1b to 1e). The effect of *dissolving transformations* increases with an increasing number of diffusion time steps $t$.

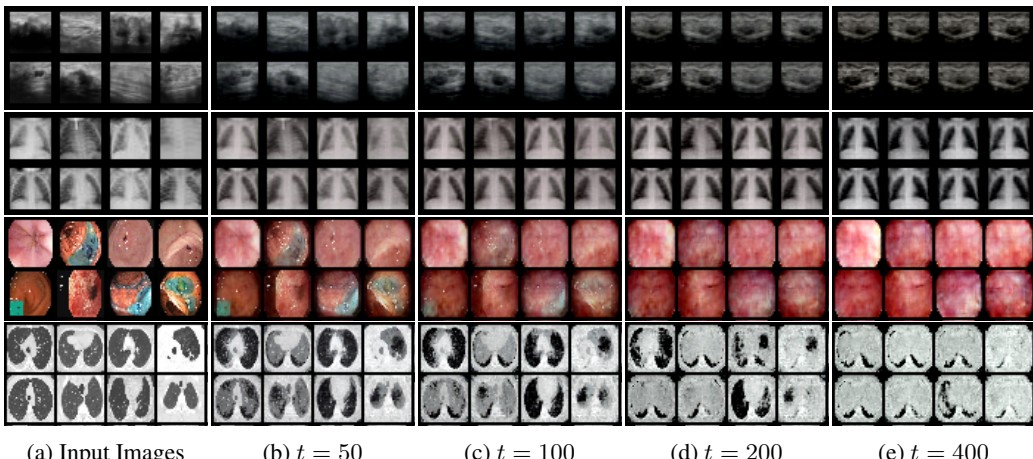

| (a) Input Images | (b) $t = 50$ | (c) $t = 100$ | (d) $t = 200$ | (e) $t = 400$ |

Figure 1: Dissolving Transformations. Figs. 1b to 1e show how the fine-grained features are dissolved (removed or suppressed). This effect is stronger as the time step $t$ is increased from left to right. In the extreme case, in Fig. 1e, different input images become very similar or almost identical depending on the dataset. We show results for four datasets from top to bottom.

To recap, diffusion models consist of forward and reverse processes, and each process is performed for $T$ time steps. The forward process $q$ gradually adds noise to an image $x_0$ for $T$ steps to obtain a pure noise image $x_T$, whereas the reverse process $p$ aims at restoring the starting image $x_0$ from $x_T$. In particular, we sample an image $x_0 \sim q(x_0)$ from a real data distribution $q(x_0)$, then add noise at each step $t$ with the forward diffusion process $q(x_t|x_{t-1})$, which can be expressed as:

$$q(x_t|x_{t-1}) = \mathcal{N}(x_t; \sqrt{1 - \beta_t} \cdot x_{t-1}, \beta_t \cdot \mathrm{I}), \tag{1}$$

$$q(x_{1:T}|x_0) = \prod_{t=1}^{T} q(x_t|x_{t-1}), \tag{2}$$

where $\beta_t$ represents a known variance schedule that follows $0 < \beta_1 < \beta_2 < \cdots < \beta_T < 1$. Afterwards, the reverse process removes noise starting at $p(x_T) = \mathcal{N}(x_T; 0, \mathrm{I})$ for $T$ steps. Let $\theta$ be the network parameters:

$$p_\theta(x_{t-1}|x_t) = \mathcal{N}(x_{t-1}; \mu_\theta(x_t, t), \Sigma_\theta(x_t, t)), \tag{3}$$

where $\mu_\theta$ and $\Sigma_\theta$ are the mean and variance conditioned on step number $t$.

The proposed *dissolving transformations* are based on Eq. (3). Instead of generating images by progressive denoising, we apply reverse diffusion in a single step directly on an input image. Essentially, we set $x_t = x$ in Eq. (3), where $x$ is the input image. We then compute an approximated state $x_0$ and denote it as $\hat{x}_{t\to0}$ to make it clear that the equation below is parameterized by the time step $t$. By reparametrizing Eq. (3), $\hat{x}_{t\to0}$ can be obtained by:

$$\hat{x}_{t\to0} = \sqrt{\frac{1}{\bar{\alpha}_t}} \cdot x - \sqrt{\frac{1}{\bar{\alpha}_t} - 1} \cdot \epsilon_\theta(x, t), \quad \bar{\alpha}_t := \Pi_{s=1}^{t}\alpha_s \text{ and } \alpha_t := 1 - \beta_t, \tag{4}$$

where $\epsilon_\theta$ is a function approximator (*e.g.* UNet) to predict the corresponding noise from $x$. Since a greater value of $t$ leads to a higher variance $\beta_t$, $\hat{x}_{t\to0}$ is expected to remove more of the "noise" if $t$ is large. In our context, we do not remove "noise" but discriminative instance features. If $t$ is small, the removed discriminative instance features are more fine-grained. If $t$ is larger, larger discriminative instance features may be removed. See Fig. 1 for examples.

### 3.2 AMPLIFYING FRAMEWORK

We propose a novel contrastive learning framework to enhance the awareness of the fine-grained image features by integrating the proposed *dissolving transformations*. In particular, we aim to enforce the model to focus on fine-grained features by emphasizing the differences between images with and without *dissolving transformations*. Fig. 2 illustrates the proposed fine-grained feature

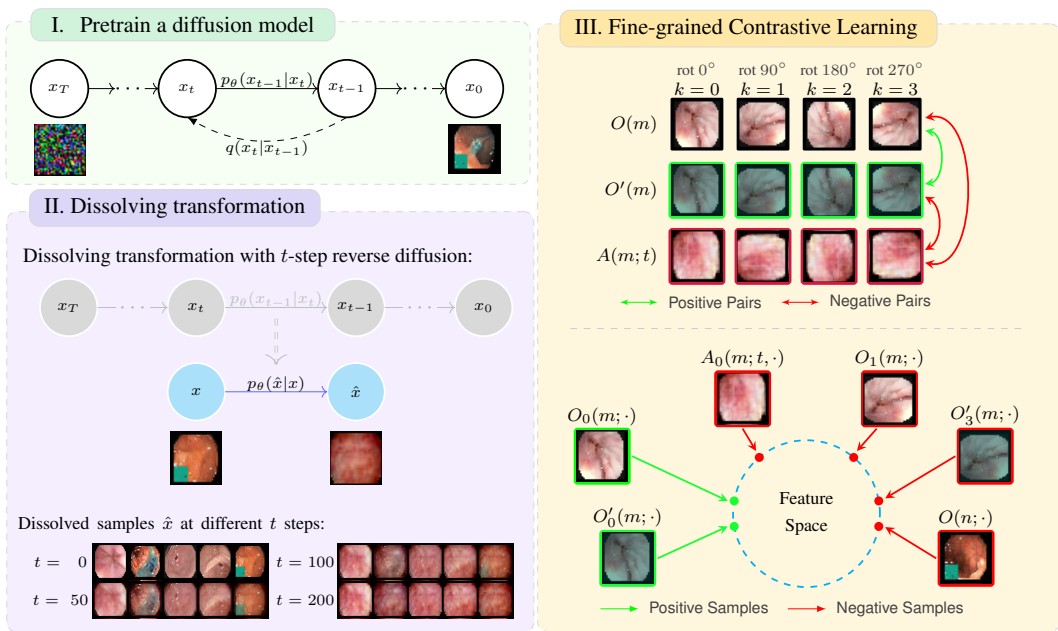

Figure 2: DIA on Kvasir-polyp dataset. (*I*) With a pretrained diffusion model, (*II*) the dissolving transformations of image $x$ result in $\hat{x}$. With a bigger time step $t$, $\hat{x}$ gradually loses fine-grained features. (*III*) Given images $m$ and $n$, we generated transformed versions $A(m; t, \cdot)$, $O(m; \cdot)$, $O'(m; \cdot)$ for image $m$ and $A(n; t, \cdot)$, $O(n; \cdot)$, $O'(n; \cdot)$ for image $n$ using a mixture of dissolving and other standard transformations. Then we form positive and negative pairs as described in Section 3.2.2. Here the two green positive samples form one positive pair. All other image pairings (between green and red or between red and red samples) form a negative pair.

learning method of DIA. We first present the different transformation branches in Section 3.2.1 and then introduce the fine-grained contrastive learning framework using our proposed *feature-amplified NT-Xent* loss in Section 3.2.2.

### 3.2.1 TRANSFORMATION BRANCHES

We extend the framework by Tack et al. (2020) for contrastive learning-based anomaly detection. They employ two types of transformations: *shifting transformations* (*e.g.* large rotations) and *non-shifting transformations* (*e.g.* color jitter, random resized crop, and grayscale). During contrastive learning, an input image is transformed by $2K$ transformations, where each transformation is the concatenation of one shifting transformation and multiple non-shifting transformations.

We use a set $\mathcal{S}$ of $K$ different *shifting transformations*. This set contains only fixed (non-random) transformations and starts from the identity $I$ so that $\mathcal{S} := \{S_0 = I, S_1, \ldots, S_{K-1}\}$. With input image $x$, we obtain $S_1(x), \ldots, S_{K-1}(x)$ as shifted images that strongly differ from the in-distribution samples $S_0(x) = x$. Each of these $K$ shifted images then passes through multiple non-shifting transformations $\in \mathcal{T}$. This yields the set of combined transformations $O := \{O_0, O_1, \ldots, O_{K-1}\}$ and $O_k = \mathcal{T} \circ S_k$. With a slight abuse of notations, we use $\mathcal{T}$ as a sequence of random non-shifting transformations. This process is then repeated a second time, yielding another transformation set $\mathcal{O}'$. We also refer to $\mathcal{O}$ and $\mathcal{O}'$ as two augmentation branches. Each image is therefore transformed $2K$ times, $K$ times in each augmentation branch. All transformations have supposedly different randomly sampled non-shifting transformations, but $O_i(x)$ and $O'_j(x)$ share the same *shifting transformation* if $i = j$.

Building on this framework, we introduce a third augmentation branch using *dissolving transformations*, denoted as $\mathcal{A} := \{A_0, \ldots, A_{K-1}\}$. The *dissolving transformations* branch outputs transformations of the form:

$$A_k = \mathcal{T} \circ S_k \circ \mathcal{D} \tag{5}$$

where $\mathcal{T}$ is a sequence of random non-shifting transformations, $S_k$ is a *shifting transformation*, and $\mathcal{D}$ is a randomly sampled *dissolving transformation*. In summary, this yields $3K$ transformations of each image, $K$ in each of the three augmentation branches.

### 3.2.2 Fine-grained Contrastive Learning

The goal of contrastive learning is to transform input images into a semantically meaningful feature representation. To design a loss function for contrastive learning, we need to decide for which of the image pairs the feature representation should be made more similar (*i.e. positive pairs*) and for which of the image pairs the feature representation should be made more different (*i.e. negative pairs*).

For a single image, we have $3K$ different transformations. In addition, we have $B$ different images in a batch, yielding $3K \cdot B$ images that are considered jointly. For all possible pairs of images, they can either be a *negative pair*, a *positive pair*, or not be considered in the loss function. We relegate the explanation to an illustration in Fig. 3. In the top left quadrant of the matrix, we can see the design choices of what constitutes a positive and a negative pair inherited from Tack et al. (2020), based on the *NT-Xent* loss Chen et al. (2020a). The region highlighted in red, is our proposed design for the new *negative pairs* for *dissolving transformations*. The purpose of these newly introduced *negative pairs* is to learn a representation that can better distinguish between fine-grained semantically meaningful features. The contrastive loss for each image sample can be computed as follows:

$$\ell_{i,j} = -\log \frac{\exp(\mathrm{sim}(z_i, z_j)/\tau)}{\sum_{k=1}^{3N} \mathbb{1}_{k,i} \cdot (\exp(\mathrm{sim}(z_i, z_k))/\tau)} \quad \mathbb{1}_{k,i} = \begin{cases} 0 & i = k, \\ 1 & otherwise, \end{cases} \quad (6)$$

where $N$ is the number of samples (*i.e.* $N = B \cdot K$), $sim(z, \hat{z}) = z \cdot \hat{z}/||z||||\hat{z}||$, and $\tau$ is a temperature hyperparameter to control the penalties of negative samples.

As mentioned previously, the positive pairs are selected from $O_i(\cdot)$ and $O'_j(\cdot)$ branches only when $i = j$. The proposed *feature-amplified NT-Xent* loss can therefore be expressed as:

$$\mathcal{L}_{con} = \frac{1}{3BK} \frac{1}{|\{x_+\}|} \sum \ell_{i,j} \cdot \begin{cases} 0 & \mathbb{1}_{i,j} \in \{x_-\} \\ 1 & \mathbb{1}_{i,j} \in \{x_+\} \end{cases}, \quad (7)$$

where $\{x_+\}$ and $\{x_-\}$ denote the positive and negative pairs, and $|\{x_+\}|$ is the number of positive pairs. The resulting target similarity matrix is shown in Fig. 3.

Additionally, an auxiliary softmax classifier $f_\theta$ is used to predict which shifting transformation is applied for a given input $x$, resulting in $p_{cls}(y^S|x)$. With the union of non-dissolving and dissolving transformed samples $\mathcal{X}_{\mathcal{S} \cup \mathcal{A}}$, the classification loss is defined as:

$$\mathcal{L}_{cls} = \frac{1}{3B} \frac{1}{K} \sum_{k=0}^{K-1} \sum_{\hat{x} \in \mathcal{X}_{\mathcal{S} \cup \mathcal{A}}} -\log p_{cls}(y^S|\hat{x}). \quad (8)$$

The final training loss is hereby defined as:

$$\mathcal{L}_{DIA} = \mathcal{L}_{con} + \gamma \cdot \mathcal{L}_{cls}, \quad (9)$$

where $\gamma$ is set to 1 in this work.

Figure 3: Visualization of the target similarity matrix ($K = 2$ with two samples in a batch). The white, blue, and lavender blocks denote the excluded, positive, and negative pairs, respectively. The red area contains the newly introduced negative pairs with dissolving transformations.

### 3.3 The Score functions

During inference, we adopt an anomaly score function that consists of two parts, (1) $s_{con}$ ensembles the anomaly scores over all shifted transformations, in addition to (2) $s_{cls}$ sums the confidence of

the shifting transformation classifier. For the $k^{th}$ shifting transformation, given an input image $x$, training example set $\{x_m\}$, and a feature extractor $c$, we have:

$$s_{con}(\tilde{x}, \{\tilde{x}_m\}) = \max_m \ \text{sim}(c(\tilde{x}_m), c(\tilde{x})) \cdot ||c(\tilde{x})||, \quad s_{cls}(\tilde{x}) = W_k f_\theta(\tilde{x}), \qquad (10)$$

$$\text{With} \quad \tilde{x} = T_k(x) \quad \tilde{x}_m = T_k(x_m)$$

where $\max_m \text{sim}(c(x_m), c(x))$ computes the *cosine similarity* between $x$ and its nearest training sample in $\{x_m\}$, $f_\theta$ is an auxiliary classifier that aims at determining if $x$ is a shifted example or not, and $W_k$ is the weight vector in the linear layer of $p_{cls}(y^S|x)$.

## 4 EXPERIMENTS

This section presents major experiment results to demonstrate the effectiveness of *DIA* for fine-grained anomaly detection.

### 4.1 EXPERIMENT SETTING

We evaluated our methods on six datasets with various imaging protocols (*e.g.* CT, OCT, endoscopy, retinal fundus) and areas (*e.g.* chest, breast, colon, eye). In particular, we experiment on low-resolution datasets of *Pnuemonia MNIST* and *Breast MNIST*, and higher resolution datasets of *SARS-COV-2*, *Kvasir-Polyp*, *Retinal-OCT*, and *APTOS-2019*. A detailed description is in Appendix A.1.

We performed semi-supervised anomaly detection that uses only the normal class for training, namely, the healthy samples. Then we output the anomaly scores for each data instance to evaluate the anomaly detection performance. We use the area under the receiver operating characteristic curve (AUROC) as the anomaly detection metric. All the presented values are computed by averaging at least three runs.

We use ResNet18 as the backbone model and a batch size of 32. In terms of *shifting transformations*, we adopted rotation as suggested by CSI (Tack et al., 2020), with a fixed $K = 4$ for $0°, 90°, 180°, 270°$. For *dissolving transformations*, all diffusion models are trained on $32 \times 32$ images. The diffusion step $t$ is randomly sampled from $t \sim U(100, 200)$ for Kvasir-Polyp and $t \sim U(30, 130)$ for the other datasets. For high-resolution datasets, we downsampled images to $32 \times 32$ for feature dissolving and then resized them back, avoiding massive computations of diffusion models.

### 4.2 TECHNICAL DETAILS

Our experiments are carried on NVIDIA A100 GPU server with CUDA 11.3 and PyTorch 1.11.0. We use a popular diffusion model implementation[2] to train diffusion models for *dissolving transformation*, and the codebase for DIA is based on the official CSI (Tack et al., 2020) implementation[3]. Additionally, we use the official implementation for all benchmark models included in the paper.

**The Training of Diffusion Models.** The diffusion models are trained with a 0.00008 learning rate, 2 step gradient accumulation, 0.995 exponential moving average decay for 25,000 steps. Each step adopts 256, 128, 32 batchsize for the resolution of $32 \times 32$, $64 \times 64$, $128 \times 128$, respectively. Adam (Kingma & Ba, 2014) optimizer and L1 loss are used for optimizing the diffusion model weights, and random horizontal flip is the only augmentation used. Notably, we found that automatic mixed precision (Micikevicius et al., 2017) cannot be used for training as it impedes the model from convergence. Commonly, the models trained for around 12,500 steps are already usable for dissolving features and training DIA.

**The Training of DIA.** The DIA models are trained with a 0.001 learning rate with cosine annealing (Loshchilov & Hutter, 2016) scheduler, and LARS (You et al., 2017) optimizer is adopted for optimizing the DIA model parameters. After sampling positive and negative samples, dissolving transformation applies then we perform data augmentation from SimCLR (Chen et al., 2020a). We randomly select 200 samples from the dataset for training each epoch and we commonly obtain the best model within 200 epochs.

---

[2]https://github.com/lucidrains/denoising-diffusion-pytorch
[3]https://github.com/alinlab/CSI

### 4.3 RESULTS

As shown in Table 1, our method beats all other methods on four out of six datasets. *RD4AD* has the best performance on two datasets. In addition, we can significantly outperform the baseline *CSI* on all datasets thereby clearly demonstrating the value of our novel components.

| Methods | | Pnuemonia MNIST | Breast MNIST | SARS-COV-2 | Kvasir-Polyp | Retinal-OCT | APTOS-2019 |
|---|---|---|---|---|---|---|---|
| KDAD (Salehi et al., 2021) | (CVPR 21) | 0.378±0.02 | 0.611±0.02 | 0.770±0.01 | 0.775±0.01 | 0.801±0.00 | 0.631±0.01 |
| †Transformly (Cohen & Avidan, 2022) | (CVPR 22) | 0.821±0.01 | 0.738±0.04 | 0.711±0.00 | 0.568±0.00 | 0.824±0.01 | 0.616±0.01 |
| RD4AD (Deng & Li, 2022) | (CVPR 22) | 0.815±0.01 | **0.759**±0.02 | 0.842±0.00 | 0.757±0.01 | **0.996**±0.00 | 0.921±0.00 |
| ‡UniAD (You et al., 2022) | (NeurIPS 22) | 0.734±0.02 | 0.624±0.01 | 0.636±0.00 | 0.724±0.03 | 0.921±0.01 | 0.874±0.00 |
| Meanshift (Reiss & Hoshen, 2021) | (AAAI 23) | 0.818±0.02 | 0.648±0.01 | 0.767±0.03 | 0.694±0.05 | 0.438±0.01 | 0.826±0.01 |
| Baseline  CSI Tack et al. (2020) | (NeurIPS 20) | 0.834±0.03 | 0.546±0.03 | 0.785±0.02 | 0.609±0.03 | 0.803±0.00 | 0.927±0.00 |
| Ours  DIA | | **0.903**±0.01 | 0.750±0.03 | **0.851**±0.03 | **0.860**±0.04 | 0.944±0.00 | **0.934**±0.00 |

†Transformaly is trained under unimodel settings as the original paper.
‡UniAD does not support $32 \times 32$ resolution. PnuemoniaMNIST and BreastMNIST datasets are trained with $128 \times 128$ resolution.

Table 1: Semi-supervised fine-grained medical anomaly detection results.

## 5 ABLATION STUDIES

This section presents a series of ablation studies to understand how our proposed method works under different configurations and parameter settings.

### 5.1 THE ROLE OF DIFFUSION MODELS

Given the challenges of acquiring additional medical data, we evaluate how diffusion models affect anomaly detection performances. Specifically, we limit the training data ratio ($\gamma$) for diffusion models to simulate less optimal diffusion models, while keeping anomaly detection settings unchanged. This experiment examines how anomaly detection performances are impacted when deployed with underperforming diffusion models with insufficient training data.

| Datasets | DIA($\gamma = 0.1$) | DIA($\gamma = 1$) |
|---|---|---|
| PneumoniaMNIST | 0.745 | 0.903 |
| Kvasir-Polyp | 0.679 | 0.732 |
| CIFAR10 | 0.916 | 0.935 |
| CIFAR100 | 0.832 | 0.883 |

Figure 4: Results with different training data ratios.

As shown in Fig. 4, a significant performance drop happened, especially for *PneumoniaMNIST*, while the other datasets also face around $0.02 \sim 0.05$ AUROC drop. Therefore, with the increasing amount of available data for training diffusion models, better performances of anomaly detection can be obtained. We conclude that the image generation quality of the diffusion model is a critical factor for obtaining decent performance.

### 5.2 ROTATE VS. PERM

Rotate and perm (*i.e.* jigsaw transformation) are reported as the most performant *shifting transformations* (Tack et al., 2020). This experiment evaluates their performances under fine-grained settings. As shown in Table 2, the rotation transformation outperforms the perm transformation.

| Method | SARS-COV-2 | Kvasir-Polyp | Retinal-OCT | APTOS-2019 |
|---|---|---|---|---|
| DIA-Perm | 0.841±0.01 | 0.840±0.01 | 0.890±0.02 | 0.926±0.00 |
| DIA-Rotate | **0.851**±0.03 | **0.860**±0.03 | **0.944**±0.01 | **0.934**±0.00 |

Table 2: Results comparison between using rotate or perm as shifting transformation methods.

### 5.3 THE RESOLUTION OF FEATURE DISSOLVED SAMPLES

This work adopted *feature-dissolved* samples with a small resolution of $32 \times 32$, which significantly improves the anomaly detection performances. Notably, the downsample-upsample routine also dissolves fine-grained features. This experiment investigates the effects of different resolutions for feature-dissolved samples. As shown in Table 3 and Table 4, the computational cost increases dramatically with increased resolutions, while it can hardly boost model performances.

| Dslv. Size | SARS-COV-2 | Kvasir-Polyp | Retinal-OCT | APTOS-2019 |
|---|---|---|---|---|
| 32 | **0.851**±0.03 | **0.860**±0.04 | **0.944**±0.01 | 0.934±0.00 |
| 64 | 0.803±0.01 | 0.721±0.01 | 0.922±0.00 | **0.937**±0.00 |
| 128 | 0.807±0.02 | 0.730±0.02 | 0.930±0.00 | 0.905±0.00 |

Table 3: Results for different resolutions for dissolving transformations.

| Res. | w/o | $32 \times 32$ | $64 \times 64$ | $128 \times 128$ |
|---|---|---|---|---|
| Params (M) | 11.2 | 19.93 | 19.93 | 19.93 |
| MACs (G) | 1.82 | 2.33 | 3.84 | 9.90 |

Table 4: Multiply–accumulate operations (MACs) for different resolutions of dissolving transformations. *w/o* denotes no dissolving transformation applied.

## 6 CONCLUSION

We proposed an intuitive *dissolving is amplifying* (DIA) method to support fine-grained discriminative feature learning for medical anomaly detection. Specifically, we introduced *dissolving transformations* that can be achieved with a pre-trained diffusion model. We use contrastive learning to enhance the difference between images that have been transformed by dissolving transformations and images that have not. Experiments show *DIA* significantly boosts performance on fine-grained medical anomaly detection without prior knowledge of anomalous features. One significant limitation is that our method requires additional training on low-resolution diffusion models for each of the datasets, instead of using the latest off-the-shelf diffusion models. We believe it would be interesting to directly adapt the SOTA off-the-shelf diffusion models to perform *dissolving transformations*. Additionally, in future work, we would like to extend our method to enhance supervised contrastive learning and fine-grained classification by leveraging the fine-grained feature learning strategy.

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
