## A APPENDIX

### A.1 DATASETS

We evaluated on benchmark *MedMNIST* datasets (Yang et al., 2021), with image sizes of $28 \times 28$:

- **PneumoniaMNIST** (Yang et al., 2021) consists of 5,856 pediatric chest X-Ray images (pneumonia vs. normal), with a ratio of $9:1$ for training and validation set.
- **BreastMNIST** (Yang et al., 2021) consists 780 breast ultrasound images (normal and benign tumor vs. malignant tumor), with a ratio of $7:1:2$ for train, validation and test set.

We also evaluated on multiple high-resolution datasets that are resized to $224 \times 224$:

- **SARS-COV-2** (Angelov & Soares, 2020) contains 1,252 CT scans that are positive for SARS-CoV-2 infection (COVID-19) and 1,230 CT scans for patients non-infected by SARS-CoV-2.
- **Kvasir-Polyp** (Pogorelov et al., 2017) consists the 8,000 endoscopic images, with a ratio of $7:3$ for training and testing. We remapped the labels to polyp and non-polyp classes.
- **Retinal OCT** (C. Basilan et al., 2023) consists 83,484 retinal optical coherence tomography (OCT) images for training, and 968 scans for testing. We remapped the diseased categories (*i.e.* CNV, DME, drusen) to the anomaly class.
- **APTOS-2019** (APTOS, 2019) consists 3,662 fundus images to measure the severity of diabetic retinopathy (DR), with a ratio of $7:3$ for training and testing. We remapped the four categories (*i.e.* normal, mild DR, moderate DR, severe DR, proliferative DR) to normal and DR classes.

### A.2 HEURISTIC ALTERNATIVES TO DISSOLVING TRANSFORMATIONS

With the proposed *dissolving transformations*, the instance-level features can hereby be emphasized and further focused. Essentially, *dissolving transformations* use diffusion models to wipe away the discriminative instance features. In this section, we evaluate our method with naïve alternatives to dissolving transformations, namely, Gaussian blur and median blur.

#### A.2.1 DIFFERENT KERNEL SIZES

We evaluate different kernel sizes for each operation. A visual comparison of those methods is provided in Fig. 5. To be consistent with the diffusion feature dissolving process, the same downsampling and upsampling processes are performed for *DIA-Gaussian* and *DIA-Median*. Referring to Table 1, though less performant, the heuristic image filtering operations can also contribute to the fine-grained anomaly detection tasks with a significant performance boost against the baseline CSI method.

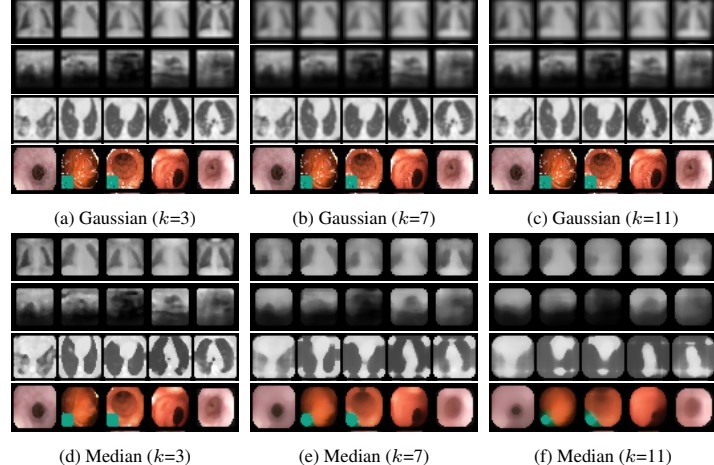

| (a) Gaussian ($k$=3) | (b) Gaussian ($k$=7) | (c) Gaussian ($k$=11) |

| (d) Median ($k$=3) | (e) Median ($k$=7) | (f) Median ($k$=11) |

Figure 5: Heuristic alternatives to dissolving transformations with various kernel sizes. Compared with median blur, Gaussian blur preserves more image semantics.

Compared against the *dissolving transformations*, those non-parametric heuristic methods dissolve image features regardless of the generic image semantics, resulting in lower performances. In a way,

| Dataset | kernel size | DIA-Gaussian | DIA-Median |
|---|---|---|---|
| pneumonia MNIST | 3 | 0.845±0.01 | 0.779±0.03 |
| | 7 | 0.839±0.04 | **0.872**±0.01 |
| | 11 | **0.856**±0.02 | 0.678±0.07 |
| breast MNIST | 3 | 0.541±0.01 | 0.641±0.03 |
| | 7 | 0.653±0.03 | **0.689**±0.01 |
| | 11 | **0.749**±0.05 | 0.542±0.04 |
| SARS-COV-2 | 3 | 0.813±0.02 | **0.837**±0.07 |
| | 7 | **0.847**±0.00 | 0.809±0.03 |
| | 11 | 0.802±0.01 | 0.793±0.02 |
| Kvasir Polyp | 3 | **0.629**±0.03 | **0.526**±0.02 |
| | 7 | 0.586±0.02 | 0.514±0.05 |
| | 11 | 0.579±0.01 | 0.495±0.04 |

Table 5: Heuristic alternatives to dissolving transformations with various kernel sizes. The blue color denotes a suboptimal performance against our proposed dissolving transformations.

*dissolving transformations* dissolve instance-level image features with an awareness of discriminative instance features, by learning from the dataset. We therefore believe that the *diffusion models* can serve as a better dissolving transformation method for fine-grained feature learning.

### A.2.2  ROTATE VS. PERM

We supplement Table 2 with the heuristic alternatives to dissolving transformations in this section. As shown in Table 6, similar to *dissolving transformations*, the rotation transformation mostly outperforms the perm transformation.

| Dataset | transform | Gaussian | Median | Diffusion |
|---|---|---|---|---|
| SARS-COV-2 | Perm | 0.788±0.01 | 0.826±0.00 | 0.841±0.01 |
| | Rotate | **0.847**±0.00 | **0.837**±0.07 | **0.851**±0.03 |
| Kvasir Polyp | Perm | 0.712±0.02 | 0.663±0.02 | 0.840±0.01 |
| | Rotate | **0.739**±0.00 | **0.687**±0.01 | **0.860**±0.03 |
| Retinal OCT | Perm | 0.754±0.01 | 0.747±0.03 | 0.890±0.02 |
| | Rotate | **0.895**±0.01 | **0.876**±0.02 | **0.944**±0.01 |
| APTOS 2019 | Perm | **0.942**±0.00 | **0.929**±0.00 | 0.926±0.00 |
| | Rotate | 0.922±0.00 | 0.918±0.00 | **0.934**±0.00 |

Table 6: Comparison between rotate and perm as shifting transformation.

### A.2.3  THE RESOLUTION OF FEATURE DISSOLVED SAMPLES

We supplement Table 3 with heuristic alternatives to dissolving transformations in this section. As shown in Table 7, those heuristic alternatives are not as performant as the proposed diffusion transformation.

| Dataset | size | DIA-Gaussian | DIA-Median | DIA-Diffusion |
|---|---|---|---|---|
| SARS-COV-2 | 32 | 0.847±0.00 | 0.837±0.07 | **0.851**±0.03 |
| | 64 | 0.821±0.01 | 0.839±0.01 | 0.803±0.01 |
| | 128 | 0.838±0.00 | 0.848±0.00 | 0.807±0.02 |
| Kvasir Polyp | 32 | 0.629±0.03 | 0.526±0.02 | **0.860**±0.04 |
| | 64 | 0.686±0.00 | 0.575±0.02 | 0.721±0.01 |
| | 128 | 0.581±0.01 | 0.564±0.02 | 0.730±0.02 |
| Retinal OCT | 32 | 0.895±0.01 | 0.876±0.02 | **0.944**±0.01 |
| | 64 | 0.894±0.00 | 0.887±0.00 | 0.922±0.00 |
| | 128 | 0.908±0.01 | 0.906±0.00 | 0.930±0.00 |
| APTOS 2019 | 32 | 0.922±0.00 | 0.918±0.00 | 0.934±0.00 |
| | 64 | 0.910±0.00 | 0.917±0.00 | **0.937**±0.00 |
| | 128 | 0.910±0.00 | 0.922±0.00 | 0.905±0.00 |

Table 7: Results for different feature dissolver resolutions.

## A.3 OTHER EXPERIMENTS

### A.3.1 NEW NEGATIVE PAIRS VS. BATCHSIZE INCREMENT

As the newly introduced *dissolving transformation* branch, given the same batch size $B$, our proposed *DIA* takes $3K \cdot B$ samples compared to the baseline *CSI* that uses $2K \cdot B$ samples. In a way, *DIA* increases the batchsize by a factor of $1.5$. Since contrastive learning can be batchsize dependent (Gutmann & Hyvärinen, 2012; He et al., 2019), we demonstrate in Table 8 that our performance improvement is not due to batch size. *CSI* with a larger batch size exhibits similar performances as the baseline *CSI* method, while the proposed *DIA* method outperformed the baselines significantly.

| Datasets | CSI | CSI-1.5 | DIA |
|---|---|---|---|
| PneumoniaMNIST | 0.834 | 0.838 | **0.903** |
| BreastMNIST | 0.546 | 0.564 | **0.750** |
| SARS-COV-2 | 0.785 | 0.804 | **0.851** |
| Kvasir-Polyp | 0.609 | 0.679 | **0.860** |

Table 8: Comparison between DIA and the batch size increment. *CSI-1.5* represents the baseline CSI models that are trained with $1.5$ times bigger batch sizes. To be specific, *CSI* and *DIA* are trained with a batch size of 32 while *CSI-1.5* used 48.

### A.3.2 THE DESIGN OF SIMILARITY MATRIX

*Shifting transformations* enlarge the internal distribution differences by introducing negative pairs where the views of the same image are strongly different. With augmentation branches $O_i$ and $O'_j$,

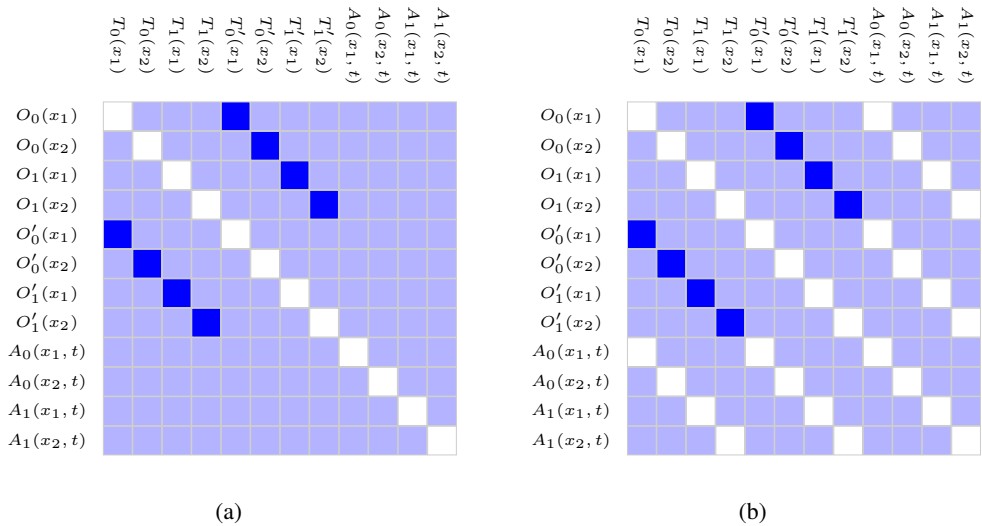

(a)                              (b)

Figure 6: Visual comparison between the similarity matrices ($K = 2$). The white, blue, and lavender blocks denote the excluded, positive, and negative values, respectively.

the target similarity matrix for contrastive learning is therefore defined where the image pairs that share the same *shift transformation* as positive while other combinations as negative, as presented in Fig. 6a. Due to the introduction of the dissolving transformation branch $A_k$, this ablation studies the design of the target similarity matrix of those newly introduced pairs. We further evaluate the design of Fig. 6b, where the target similarity matrix is designed to exclude the image pairs with and without dissolving transformations applied whilst sharing the same *shift transformation*, when $i = k$ or $j = k$. Essentially, these pairs share the same *shift transformation* which should be considered as positive samples, but the $A_k$ branch removes features that make them appear negative. Thus, we investigate whether these contradictory samples should be considered during the contrastive learning process.

| Methods | | SARS-COV-2 | Kvasir-Polyp | Retinal-OCT | APTOS-2019 |
|---|---|---|---|---|---|
| Baseline | CSI | 0.785 | 0.609 | 0.803 | 0.927 |
| Ours | DIA-(a) | 0.851 | 0.860 | 0.944 | 0.934 |
| Ours | DIA-(b) | 0.850 | 0.843 | 0.932 | 0.930 |

Table 9: Semi-supervised fine-grained medical anomaly detection results.

As shown in Table 9, those designs achieve very similar performances on medical datasets. Then, we further evaluate our methods on standard anomaly detection datasets, that contain coarse-grained feature differences (*i.e.* Car vs. Plane) with a minimum need to discover fine-grained features.

| Dataset | Method | | 0 | 1 | 2 | 3 | 4 | 5 | 6 | 7 | 8 | 9 | avg. |
|---|---|---|---|---|---|---|---|---|---|---|---|---|---|
| | Baseline | CSI | 89.9 | 99.1 | 93.1 | 86.4 | 93.9 | 93.2 | 95.1 | 98.7 | 97.9 | 95.5 | 94.3 |
| CIFAR10 | Ours | DIA-(a) | 90.4 | 99.0 | 91.8 | 82.7 | 93.8 | 91.7 | 94.7 | 98.4 | 97.2 | 95.6 | 93.5 |
| | Ours | DIA-(b) | 80.0 | 98.9 | 80.1 | 74.0 | 81.2 | 84.4 | 82.7 | 94.7 | 93.9 | 89.7 | 86.0 |

| Dataset | Method | | 0 | 1 | 2 | 3 | 4 | 5 | 6 | 7 | 8 | 9 | |
|---|---|---|---|---|---|---|---|---|---|---|---|---|---|
| | Baseline | CSI | 86.3 | 84.8 | 88.9 | 85.7 | 93.7 | 81.9 | 91.8 | 83.9 | 91.6 | 95.0 | |
| | Ours | DIA-(a) | 85.9 | 82.6 | 87.0 | 84.7 | 91.8 | 84.4 | 92.1 | 79.9 | 90.8 | 95.3 | |
| | Ours | DIA-(b) | 83.2 | 80.4 | 86.1 | 83.0 | 90.8 | 78.2 | 90.6 | 75.8 | 86.7 | 92.5 | |
| CIFAR100 | Method | | 10 | 11 | 12 | 13 | 14 | 15 | 16 | 17 | 18 | 19 | avg. |
| | Baseline | CSI | 94.0 | 90.1 | 90.3 | 81.5 | 94.4 | 85.6 | 83.0 | 97.5 | 95.9 | 95.2 | 89.6 |
| | Ours | DIA-(a) | 93.0 | 90.1 | 89.9 | 76.7 | 93.1 | 81.7 | 79.7 | 96.0 | 96.3 | 95.2 | 88.3 |
| | Ours | DIA-(b) | 91.2 | 86.3 | 87.7 | 73.3 | 91.8 | 80.7 | 79.7 | 97.2 | 95.3 | 93.3 | 86.2 |

Table 10: Results on standard benchmark datasets. Results are AUROC scores that are scaled by 100.

We therefore include the following datasets:

**CIFAR-10** consists of 60,000 32x32 color images in 10 equally distributed classes with 6,000 images per class, including 5,000 training images and 1,000 test images.

**CIFAR-100** similar to CIFAR-10, except with 100 classes containing 600 images each. There are 500 training images and 100 testing images per class. The 100 classes in the dataset are grouped into 20 superclasses. Each image comes with a "fine" label (the class to which it belongs) and a "coarse" label (the superclass to which it belongs), which we use in the experiments.

Note that the corresponding diffusion models for each experiment are trained on the full CIFAR10 and CIFAR100 datasets, respectively.

As shown in Table 9 and Table 10, the exclusion of the $i = k$ and $j = k$ pairs barely affect the performance for the fine-grained anomaly detection tasks, but significantly lowers the performance for the coarse-grained anomaly detection tasks.