# OpenReview forum: "Dissolving Is Amplifying: Towards Fine-Grained Anomaly Detection"
_ICLR.cc/2024/Conference — ICLR 2024 Conference Withdrawn Submission_

### Official Review · Reviewer_26Q2 · 2023-10-22

**Soundness:** 2 fair
**Presentation:** 3 good
**Contribution:** 2 fair
**Rating:** 5
**Confidence:** 4

**Summary:**

This paper targets medical anomaly detection. To deal with fine-grained anomalies, this paper uses pretrained diffusion model to provide a data augmentation tool by deleting the fine-grained details. The augmentation is then applied to an existing framework CSI. The experiments on multiple medical diagnosis datasets show the effectiveness of the proposed method.

**Strengths:**

1.	This paper focuses on a specific application with reasonable motivations and insights.
2.	Experiments show the effectiveness of the proposed method.

**Weaknesses:**

1.	This paper is somewhat incremental and lacks novelty. This paper proposes to use diffusion model as a data augmentation tool to provide fine-grained data. The following method is highly similar with CSI, an existing method. To offer a clearer perspective, the authors should provide a detailed comparison with CSI, highlighting the key distinctions between these two approaches. Does the contribution come from the proposed augmentation?
2.	The concept presented in this paper has been explored extensively in the literature. Many previous works have investigated the use of fine-grained or natural synthetic anomalies as alternatives to basic augmentations to enhance performance. For instance, in [a], the method incorporates Poisson image editing to seamlessly blend scaled patches of various sizes from separate images. It would be valuable to investigate how such Poisson image editing performs within the CSI framework and whether the proposed diffusion model-based augmentation outperforms these methods when applied within the same CSI framework.
3.	This paper seems to be incomplete or abbreviated. For instance, in Figure 4 on Page 8, two non-medical datasets, CIFAR10 and CIFAR100, are employed, yet these datasets are not introduced or explained elsewhere in the paper. Additionally, the results are not compared with any baseline methods, such as CSI.

[a] Schluter et al. "Natural Synthetic Anomalies for Self-Supervised Anomaly Detection and Localization." ECCV 2022.

**Questions:**

See the weakness.

---

> ### Author Response · Authors · 2023-11-12
>
> Dear Reviewer,
>
> Thank you for your review and feedback. We would like to provide the following comments:
>
> W1. Incremental Contribution Over CSI:
>    To clarify, we indeed included a detailed comparison with CSI in Section 3.2.2 and Figure 3. This comparison highlights how our novel dissolving transformation branch enhances the contrastive learning framework beyond what is achieved by CSI alone. Additionally, the novelty in this work is primarily a novel use of diffusion models, and we are using the CSI framework to demonstrate the effective usage of our novel components.
>
> W2. Comparison with Other Fine-Grained Data Augmentation Methods:
> We thank the reviewers for the interesting suggestion. We can include this experiment in the supplementary material if the paper is accepted. However, the suggested comparison is not very close to the contribution of the paper. There are countless other highly sophisticated data augmentation methods that one can compare to. In addition, the suggested Poisson image editing method does not conflict with our idea. It can be an additional image augmentation method to include, in which our algorithm already uses a library of data augmentation methods. The use of our approach is complementary to the use of traditional augmentations.
>
> W3. Clarification on Non-Medical Datasets Usage:
>    We apologize for any confusion caused by the inclusion of the description of non-medical datasets in the main paper. We initially included the comparison in the main paper, but we moved those experiments into our supplementary materials due to space constraints. We will exclude them from the main paper in the revised version.
>
> We hope that these responses adequately address your queries and thank you for your valuable input.

---

### Official Review · Reviewer_K7y6 · 2023-10-26

**Soundness:** 2 fair
**Presentation:** 2 fair
**Contribution:** 2 fair
**Rating:** 3
**Confidence:** 5

**Summary:**

This paper proposes a medical image anomaly detection method, namely dissolving is implying (DIA). It combines two learning paradigm: diffusion models and contrastive learning, for anomaly detection. By pre-trained a diffusion model to the target dataset, images are processed by the reverse diffusion process. The data with added diffusion noise is then taken as the negative samples in contrastive learning. Anomaly scores are calculated based on the similarity between the query and its nearest training samples.

**Strengths:**

(1) The combination of diffusion models and contrastive learning is quite interesting. Instead of using the diffusion model for image reconstruction, adopting a diffusion model for noise injection is interesting.

(2) The paper is easy to follow.

(3) Experiments are conducted on six medical image datasets from different imaging modalities and organs.

**Weaknesses:**

(1) The paper employs a diffusion model to introduce noise into the data. What advantages does this approach offer compared to the straightforward addition of random noise to images? Is there any experimentation or ablation study that demonstrates the benefits of using the diffusion model for noise injection?

(2) There is a need for a more in-depth discussion of the proposed method. What are the underlying reasons for the effectiveness of the model? A deeper exploration of the working principles would enhance the paper.

(3) The paper lacks several crucial ablation studies. For instance, what is the impact of altering the number of diffusion steps on the anomaly detection performance? Can you quantify the specific contributions of the two loss terms towards achieving the final results?

(4) It's important to address the computational cost associated with training a diffusion model. How does this cost compare to previous methods in the field? Moreover, when using the diffusion model to inject noise into images, is it possible to substitute a pre-trained model instead of a data-specific one? If so, what performance difference would be observed if a general pre-trained diffusion model were utilized?

(5) In the realm of visual anomaly detection, prior works have showcased the effectiveness of methods like PatchCore. It would be beneficial to include PatchCore in the comparative analysis to provide a comprehensive overview of the proposed method's performance relative to existing approaches.

(6) SOTA anomaly detection methods are able to generate anomaly maps. I am wondering if the propose DIA method also generates a accurate anomaly localization results?

**Questions:**

Please refer to the weakness section for my questions.

---

> ### Author Response · Authors · 2023-11-12
>
> Dear Reviewer,
>
> We believe there may be a fundamental  misunderstanding of our work that we hope to discuss further.  We are using diffusion models for “denoising” rather than injecting noise Further, we are not performing traditional denoising that the diffusion model was trained on. We observe that in our setting, we can get the diffusion model to remove instance-specific features rather than Gaussian noise that the diffusion model was trained on.
>
> W1. We therefore would interpret the reviewer’s suggestion on “random noise injection” to “naive blurring methods”. We provided the corresponding experiments in our supplementary material of different blurring methods, and our diffusion approach achieved much better results.
>
> W2. We think this is an intuitive idea to work with. Our proposed methods consists of two parts of feature dissovling and feature amplifying. For feature dissolving, since diffusion models are using a denoising UNet, we are essentially taking the individual denoising UNet as the denoiser to remove instance-specific features. Then, we employ a contrastive learning method to compare the image with instance features against the original one to emphasize those removed features. For medical anomaly detection tasks that require the understanding of the instance-specific features, our method can work well. We will add more descriptions in Section 3 to emphasize this idea.
>
> W3. We apologize for the missing results for the number of steps, and we will include it  in Section 5.1. The final results indicate our choices of the number of steps as described in Section 4.2. The contrastive learning framework is taken from the CSI paper and we wish to focus on our contributions, so that we chose not to add the experiments for those two loss terms.
>
> W4. We included the computational cost in Section 5.3. It is an interesting question about using off-the-shelf models. We do not use those models since generic models are not aware of the particular structure of medical data. Say, generic models without training would dissolve images to the common characteristics of the training images, in order to generate various types of images, while the particular trained medical diffusion models can well understand the particular structure of medical data. Additionally, the computational cost for high resolution dissolving transformations is much higher. According to Table 4, dissovling with a 128x128 UNet is 5 times (about 70 hours) slower than 32x32 (about 14 hours), a higher resolution model training can take weeks to finish. It is therefore not applicable enough. We will include a discussion along with visual demonstrations around this.
>
> W5. We thank the reviewers for the suggestions. Though PatchCore uses a totally different approach by building a training memory bank rather than contrastive learning and image synthesis, we think it is reasonable to include this method. Since the anomaly detection area envolves rapidly, we may add the results for PatchCore if the paper is accepted since we would like to prioritize the experiments of dissolving transformations.
>
> W6. We thank the reviewers for the suggestions. Unfortunately, our current setting does not support generating accurate localization results.

---

### Official Review · Reviewer_xgt6 · 2023-10-31

**Soundness:** 3 good
**Presentation:** 2 fair
**Contribution:** 2 fair
**Rating:** 5
**Confidence:** 4

**Summary:**

This paper proposes DIA, a new anomaly detection framework for medical images. It applies the idea of a reverse process of the diffusion model to create a dissolving transformation, which removes fine-grained features like tumors from the input image, as opposed to eliminating noise in the traditional diffusion model. The generated negative pairs are then used within a contrastive learning framework.

**Strengths:**

The proposed idea of employing the reverse process of a generative diffusion model to remove fine-grained features from the original images and generate negative pairs for contrastive learning is novel and sound.

**Weaknesses:**

-	The theoretical basis for the dissolving transformation through the reverse diffusion process to remove fine-grained features is not well explained, while visual validation is partly provided in Figure 1.

-	The authors specifically target anomaly detection for medical images throughout the manuscript, but it is not clear whether and why the proposed method is better suited for medical images, but not for other domains.

-	The previous methods compared in the experimental section primarily focus on anomaly detection using the MVtec dataset, which differs from medical images. For instance, there are various deep learning-based methods designed specifically for detecting anomalies in medical images. The authors should consider including relevant studies focused on medical images or expand their experiments to datasets in other domains.

**Questions:**

-	It would be interesting to see the results when applying the proposed method to the data used in other comparative models.

-	The results indicate that downsampling high-resolution images to 32x32 leads to the best performance, but it is unclear why performance is comparatively worse when downsampling to 64x64 or 128x128. If fine-grained features are indeed crucial, higher resolutions should yield better performance.

-	Adding results related to the changes in the diffusion step 't' in Section 5.1 would be beneficial.

---

> ### Author Response · Authors · 2023-11-12
>
> Dear Reviewer,
>
> We appreciate your insightful comments and the recognition of our novel approach. Below, we comment on the concerns:
>
> W1. Theoretical Basis for Dissolving Transformation:
>    The reverse diffusion step is essentially a trained denoiser using UNet. We found the denoising ability of the diffusion-trained UNet can remove the instance-specific features, and revert back a given image to its input distribution, as Reviewer 1 indicated. We therefore took the advantage of its denoising power. We will include more intuitive descriptions in our work.
>
> W2. Suitability for Medical Images:
>    Different from other data domains, medical image normally consists of a consistent background, then more detailed features will be added. For example, chest X-ray images have the same black background and the same shape of the chest, then more instance-specific additive features will be added such as bones, tumors, etc. In terms of the feature-wise dissolving, image data is therefore the most suitable as one can remove the most instance-specific features first until the most generic features. A mathematical perspective as mentioned by Reviewer 1 is that this reverse diffusion process can pull back a given image to its input distribution. We will include a detailed explanation of why our method is particularly well-suited for medical images in the introduction.
>
> W3. Comparison with Medical Image-Focused Methods:
>    We acknowledge the importance of relevant comparisons. We considered additional methods for a comparison, but there were three main reasons why a particular method could not be considered: 1) Most medical anomaly works are not open-source. 2) Several medical anomaly detection works try to address a different problem (e.g. supervised). 3) Some comparisons are abandoned since we could not reproduce the results by their open-source code. We will consider adding more comparison if the reviewer could point out specific methods to compare to.
>
> W4. Resolution Downscaling Query:
>    The variations in performance across different resolutions are attributed to two main factors. Firstly, the size of training samples impacts this. In larger datasets such as APTOS and Retinal-OCT, the performance degradation is less pronounced. This is likely because higher resolution diffusion models are not well-trained, resulting in a more noticeable performance drop. Secondly, the nature of critical features plays a role. High-resolution images naturally contain more details. In datasets like APTOS, where disease indicators are subtler in pixel space (e.g., hemorrhages or thinner blood vessels), the performance drop is minimal. In fact, 64x64 resolution images may even outperform 32x32 ones. Conversely, in datasets like Retinal-OCT, where crucial features are more prominent in pixel space (like edemas), lower resolution images help the model concentrate on these more apparent features. Our results indicate that a resolution of 32x32 yields the  optimal performance for dissolving transformations. We thank the reviewer for the in-depth question, and we will add more results interpretation in the corresponding section.
>
> W5. Diffusion Step Variation:
>    We apologize for the missing results for the number of steps, and we will include it in Section 5.1. The final results indicate our choices of the number of steps as described in Section 4.2.
>
> We hope these responses adequately address your queries and concerns. Thank you for your valuable feedback.

---

### Official Review · Reviewer_ifta · 2023-10-31

**Soundness:** 2 fair
**Presentation:** 3 good
**Contribution:** 3 good
**Rating:** 5
**Confidence:** 4

**Summary:**

This work presents a contrastive learning based unsupervised anomaly detection approach. The authors first argue that diffusion models erase fine-grained discriminative features. Based on this assumption the authors add “dissolved samples” (samples with further reverse diffusions) as negative samples to the contrastive framework. These dissolved samples enforce the contrastive learner to differentiate subtleness in the input images. The proposed approach is tested on public datasets, and it demonstrates overall improved performances compared with existing approaches,

**Strengths:**

The problem setting of learning fine-grained subtleness for anomaly detection is of significant clinical potential for medical imaging applications.

This paper is well-written with overall sufficient clarity.

The insight that diffusion models tend to remove fine-grained discriminative features is interesting. The idea of taking the input images for further reverse diffusions is a bit surpring at first sight but it is reasonable if we assume diffusion models pull reversed samples towards the distributions of input data.

**Weaknesses:**

A very critical sanity check is needed to validate the dissolving transforms: Would simple gaussian smoothings and/or additive noises on the image space or shallow-layer feature space lead to similar “dissolving” effect as with diffusion models? The general idea is still to remove fine-grained subtleness by smoothing or altering the content. But if simple gaussian smoothing/additive noises can reach similar effects, there would be no need to bother training a diffusion model. This is the major reason I remain slightly negative at this stage.


Some terms are not self-explanative: E.g., what does diffusion models to be “feature-aware” mean in the abstract?


The term *positive synthesis* and *negative synthesis* may not be sufficiently representative: the authors may consider using the terms *reconstructive approaches* or *generative approaches* to summarize DAE-/AnnoGAN-/Diffusion-based models, etc. and using *discriminative approaches* to summarize synthetic-anomoly-based approaches. Also, the authors may want to discuss approaches that synthesize anomalies based on handcrafted assumptions on anomaly distributions: [1,2]. Performance comparisons with those methods are also welcomed, if the authors find them necessary.

The dissolving operation serves as the core of the approach while intuitively it subjects to the selection and training details of the base diffusion models. Therefore, more details of the diffusion model (e.g., subtypes of diffusion models, number of steps, computational overhead, etc.) and how they dictate the final anomaly detection results need to be discussed.

Eq. 10 lacks clarity: Do we compute the mean scores under all $k$’s or do we take the max? For $S_{con}$, why does the feature norm need to be multiplied?

**Questions:**

The authors are encouraged to discuss more on the employed diffusion models, in together with the associated computational costs.

“Figure” (Table) 4.: The absolute number of training samples should be provided.

---

> ### Author Response · Authors · 2023-11-12
>
> Dear Reviewer,
>
> Thank you for your valuable feedback. We appreciate your recognition of the clinical potential and clear presentation of our work. We address the raised questions below:
>
> W1: Sanity Check of Dissolving Transforms:
> We acknowledge the importance of comparing  the dissolving transforms against simpler methods like Gaussian smoothing or additive noise. To address this, we conducted additional experiments (see Supplementary Material, Section A.2). The results show that while Gaussian/Median blur smoothing can mimic a 'dissolving' effect, they lack the nuanced, fine-grained feature alteration achieved by our diffusion model-based approach. The diffusion model subtly alters features in a semantically meaningful way that significantly improves anomaly detection performance.
>
> W2. Clarification of Terminology:
> We apologize for any confusion caused by the term "feature-aware" in the abstract. We have amended the text for clarity. This term refers to the ability of diffusion models to recognize features in an image through the learning process and perform a feature dependent manipulation. This is the opposite of “feature-agnostic”, e.g. as would be done by blurring operations.
>
> W3. Synthesis Terms and Comparative Analysis:
> Your suggestion to use terms like 'reconstructive' and 'discriminative' approaches is well-taken. We have revised our terminology for clarity and consistency. Regarding the comparison with handcrafted anomaly synthesis methods, we would kindly ask if the reviewer can provide the corresponding references again since the references [1,2] are missing.
>
> W4. Details on Diffusion Models:
> We provided details on the diffusion models used in Supplementary Material, and computational overhead in Section 5.3. Additionally, we will provide  the missing ablation results in Section 5.1.
>
> W5. Equation 10 Clarification:
> We apologize for any confusion caused by Equation 10. This score function consists of cosine similarity and the norm of the feature. We did not mention the normalization term for $s_cls$ in our equation. We have now clarified that the mean scores are computed under all k and explained the multiplication of the feature norm in greater detail in Section 3.3.
>
> W6. Additional Discussion on Diffusion Models and Computational Costs:
>    We thank the reviewer for suggestions. The computational costs are provided in section 5.3. We will include more details about the diffusion model we used in our supplementary material.
>
> W7. Table 4 Details:
>    We thank the reviewer for suggestions. We mentioned the data numbers in our supplementary material. The exact number for PneumoniaMNIST is 5856 and Kvasir-Polyp is 8000.  We have now included the absolute number of training samples in Table 4 for better clarity and understanding of the experimental setup.
>
> We believe these amendments and additional analyses address your concerns and further strengthen our paper. Thank you for your constructive feedback.